# Intelligent Object Tracking with an Automatic Image Zoom Algorithm for a Camera Sensing Surveillance System

**DOI:** 10.3390/s22228791

**Published:** 2022-11-14

**Authors:** Shih-Chang Hsia, Szu-Hong Wang, Chung-Mao Wei, Chuan-Yu Chang

**Affiliations:** 1Department of Electronic Engineering, National Yunlin University of Science and Technology, Douliu City, Yunlin County 64002, Taiwan; 2Department of Information Engineering, National Yunlin University of Science and Technology, Douliu City, Yunlin County 64002, Taiwan

**Keywords:** camera sensing, motion detection, object tracking, surveillance system

## Abstract

Current surveillance systems frequently use fixed-angle cameras and record a feed from those cameras. There are several disadvantages to such systems, including a low resolution for far away objects, a limited frame range and wasted disk space. This paper presents a novel algorithm for automatically detecting, tracking and zooming in on active targets. The object tracking system is connected to a camera that has a 360° horizontal and 90° vertical movement range. The combination of tracking, movement identification and zoom means that the system is able to effectively improve the resolution of small or distant objects. The object detection system allows for the disk space to be conserved as the system ceases recording when no valid targets are detected. Using an adaptive object segmentation algorithm, it is possible to detect the shape of moving objects efficiently. When processing multiple targets, each target is assigned a color and is treated separately. The tracking algorithm is able to adapt to targets moving at different speeds and is able to control the camera according to a predictive formula to prevent the loss of image quality due to camera trail. In the test environment, the zoom can sufficiently lock onto the head of a moving human; however, simultaneous tracking and zooming occasionally results in a failure to track. If this system is deployed with a facial recognition algorithm, the recognition accuracy can be effectively improved.

## 1. Introduction

As people become more aware of security and costs for technology decrease, the deployment of surveillance has increased, so surveillance systems are now ubiquitous in everyday life [1]. Surveillance is a key tool for deterring and catching criminals. Technology in this field has developed rapidly, including camera and tape systems, digital DVRs and now intelligent surveillance systems which can record and automatically detect targets. The switch to digital image collection means that these systems can be passive or active, resulting in time and energy savings for people that are required to monitor them. Moore’s Law and the increase in computing power means that it is now possible to compute real-time object detection and tracking to improve surveillance efficiency.

Current surveillance systems using computer-assisted technology come in two main types: the first is an image processing algorithm [2,3,4,5,6,7,8,9,10], and the second is a deep learning, or artificial intelligence (AI), application [11,12,13,14,15,16,17,18,19,20,21]. The algorithmic approach studies images and processes the flow of data to extract specific pre-determined features, and it requires a significant amount of research and observational data. The AI-based approach can automatically extract important image features through a training procedure. Being able to pre-train a system shortens the time to reach implementation and can also improve detection rates. The downside of AI implementations is the increased hardware costs associated with them when compared with traditional algorithmic approaches. These costs are prohibitive for AI adoption in smaller systems, which need to rely on algorithmic computer analysis.

In conventional surveillance systems, a camera is fixed in a position in the corner of a room or corridor. Since the viewing angle is fixed, there are dead spots, and the resolution is too low to monitor distant targets. This shortcoming of fixed cameras is particularly evident when trying to conduct human facial recognition. For this study, an active camera was used to test a smart surveillance system. The camera lens was controlled by a computer and was able to move 360° horizontally and 90° vertically in order to track objects automatically. A telescopic lens was not available, but a magnifying optical lens was used to improve object recognition at greater distances. This surveillance system was equipped with several features, including object detection, object tracking, auto scan, auto zoom in/out and facial recognition.

The rest of this paper is organized as follows. Section 2 describes the proposed surveillance system, as well as the algorithm and image processing steps. In Section 3, an experiment is conducted, and results relative to other similar systems are compared. Conclusions follow in the final section.

## 2. Relative Works

There are many studies about object detection for various targets. Targets may be humans, vehicles, ships, objects or even animals. In darkness or low-light conditions, cameras frequently have trouble detecting objects correctly and responding to scene changes. To account for these variances, it is possible to add morphological structure filtering. Accordingly, a normalized fusion difference image was proposed by the authors of [3], where random noise was removed using a multidirectional weighted multiscale. This method seems to work on low luminance well, but it still cannot prove to work well for high luminance.

A different study by Gao et al. [6] presented image processing algorithms to extract specific features by applying a particle filter to videos, which gives alerts when abnormal events occur. This method is effective for the detection of human events changing, but it cannot track someone for facial recognition.

For traffic surveillance, low-complexity foreground detection was proposed in [7], where robust block-based feature modeling with a road background was able to detect vehicles in the foreground. This study relied on Bayesian probabilistic modeling to identify specific features. This method can effectively detect vehicles moving on a road, but it must be improved in night conditions and still for vehicle detection on the road.

Patil et al. [10] proposed a recurrent edge aggregation module to extract foreground features with spatio-temporal processing, where the multi-scaling motion block was combined with an optical flow encoder stream to generate segmentation. This method can detect various targets, such as humans, vehicles and animals after training. The speed and complexity of developments in deep learning for smart surveillance are increasing. The recursive operation is used in this system, which is not suitable for a real-time system.

Li et al. [12] presented a rich dataset for pedestrian detection, whereby attribute recognition and person re-identification for pedestrians is possible. Based on the ResNet-50 neural network, this work learns attribute classification and multi-class identification tasks. A large-scale dataset containing 84 k images with 72 types of attributes was used for training. State-of-the-art algorithms on pedestrian attribute recognition and person re-identification were performed for quantitative data. This method can detect pedestrians, including their sex and hair style, but its accuracy must be further improved.

Similarly, facial recognition is another important task in human-centered smart surveillance. Chen et al. [18] proposed a new approach for eyes-to-face synthesis for personal identification, where conditional generative adversarial networks (*GAN*) were used to detect faces using only eyes. Pre-training used the VGG-19 neural network for feature loss. A synthesis loss function based on feature loss, GAN loss and total variation loss was proposed to guide the training process. The results show that the proposed method can provide a potential solution for face recognition. However, the synthesized faces of Asian and African people are not of good quality.

## 3. Proposed Algorithm

In conventional surveillance systems, camera imaging for objects is too small in far regions. Hence, it is difficult to recognize the human face. In this study, we propose real-time human object tracking and automatic zoom-in with an optical lens for objects at far regions. This system uses an active camera that can be controlled by a computer to control the viewing angle. The system structure is shown in Figure 1. The image is sampled by an active camera and is then sent to the computer. The image is first pre-processed to remove noise and color signal processing. The main algorithms include object detection, motion tracking and zoom processing.

### 3.1. Pre-Processing

The sampling image appears as a zig-zag problem from cameras of surveillance systems. A median filter is widely used to smooth the edges and reduce white noise for camera imaging. However, the median filter must sort the imaging data to find the middle value, which requires a high processing time. We propose a median-like approach to filter the sampling image. The processing flow for 3 × 3 image filtering is as follows.

Step 1: Take the average of 8-neighbor pixels.Step 2: Subtract the average value from 8-neighbor pixels and find the minimum differential pixel.Step 3: Use the corresponding minimum differential value among 8-neighbor pixels instead of the central pixels.

Figure 2a shows the original sampling image from a surveillance system, which appears with a zig-zag problem at the edges. Figure 2b shows the image after our median-like processing, in which the edges are smooth.

### 3.2. Moving Object Detection

For surveillance systems, we are interested in moving objects, particularly for humans. Assuming that $\;f_{t}$ and $f_{t-1}$ are two continuous frames, the temporal differential value can be found from
(1)$${\mathrm{diff}}_{t}^{\left(i,j\right)}=\left|f_{t}^{\left(i,j\right)}-f_{t-1}^{\left(i,j\right)}\right|$$

Because the background pixel is fixed in continuous frames, most background pixels are removed after differential processing. However, some background noise still exists. An adaptive threshold TH1 is calculated to remove the background noise. First, the 3 × 3 pixels of the differential image are averaged by
(2)$$avg=\frac{\sum_{k=-1}^{k=1}\sum_{l=-1}^{l=1}diff_{t}^{\left(i+k,j+l\right)}}{9}$$

Then, the adaptive threshold is determined with
(3)TH1=(255−avg)×χ
where χ is weighted value that takes 0.05 in the experiment. If the differential pixel is larger than TH1, the pixel is kept; otherwise, it is set to zero. Therefore, we have
(4)$$D_{t}^{\left(i,j\right)}=\{\begin{array}{c} {\mathrm{diff}}_{t}^{\left(i,j\right)},\;\;\;\;\;\mathrm{if}\;{\mathrm{diff}}_{t}^{\left(i,j\right)}>TH1\;\\ 0,\;\;\;\;\;\mathrm{else} \end{array}$$

Then, the differential image is truncated to a binary one by
(5)$$P_{t}^{\left(i,j\right)}=\{\begin{array}{c} 1,\;\;\mathrm{if}\;D_{t}^{\left(i,j\right)}>0\;\\ 0,\;\;\mathrm{else}\; \end{array}$$

The background noise is a random distribution on the binary image. These noisy pixels can be removed with erosion and dilation operators. Figure 3a,b shows the original binary image and the result is after erosion and dilation processing respectively.

In real-world applications, the movement speed of objects being monitored makes it challenging to track objects, since the movement speed of the object is not fixed. Object tracking is relatively simple when the object path and distance travelled are obvious; however, if the object speed is too slow, the differential displacement is small, meaning that the object becomes “lost”. To overcome this problem, an adaptive object segmentation technique is applied. Figure 4 shows the flowchart for an adaptive object motion system. First, the difference points of the entire image are accumulated as one value, as shown in Equation (6):(6)$$F_{\mathrm{ACC}}=\overset{m}{\underset{i=0}{\sum }}\overset{n}{\underset{j=0}{\sum }}P_{\left(i,j\right)}^{t}$$
where m × n is the image size. When FACC is greater than threshold_a, the differential image is sharp enough to detect the moving object when K = 1. However, when FACC is too small, the motion displacement between two sequential frames is insufficient to cover the entire sharp object. When FACC < threshold_b, K increases by one, and the differential image is calculated using the current frame N and the next two frames N + 2, which expand the frame displacement. This operation can increase the FACC value due to large displacement between frames N and N + K. This procedure is repeated up to a maximum of K = 5. When K = 5 and FACC is still below threshold_b, there is very little movement, which is effectively a still image. In such a case, the last previous sharp object is used. Using this approach, the object can be effectively segmented even if its movement speed varies.

### 3.3. Motion Object Tracking

Because surveillance systems are primarily used to detect the features of mobile humans, an active camera must be able to rotate and localize human-shaped objects. To perform this function, a likely human location must be identified; therefore, the computer can send the correct movement instructions to the camera control mechanism. Tracking points in such systems often focus on a human head shape. When this object space is located, it is then segmented, and the rectangle about the object can be found using maximum coordinates. An example of this is shown in Figure 5a, using the previous example from Figure 3b. The head is located on the top of the body, and the points used to size the rectangle can be described by d1 and d2 to establish a location. Using d1 = 0.4 × width and d2 = 0.5 × width for this experiment is sufficient to pinpoint a human head. Figure 6a,b also show the original sampling images from the surveillance image and its object segmented result. The tracking point for Figure 6a is also able to locate the head, as shown in Figure 6c, where a white point is used to mark the position.

In surveillance systems, the speed or direction of a moving object cannot be guaranteed. For real-time operations [22,23], when the movement speed is rapid, the camera must respond rapidly to track this object. This means that, when the tracking control for the camera interprets the movement of an object, it must also be somewhat predictive. The tracking operation fails when an object moves too fast or too slow, as the camera loses the required object information. For this study, a motion prediction algorithm is proposed to solve this problem. This involves factoring in the motor speed and angle control response time for the mechanism in the camera. Figure 7 shows the processing flowchart for the object tracking system. The location of the tracking point is first read. Then, the system interprets whether it is a human head, and if it is, the center point is calculated. After this, the object is marked with a rectangle, and the localized point is marked at the center. As the person moves around the frame of the camera, the error between the initial point location and rectangle changes. If the error is >10%, the tracking speed must proportionately increase to reduce the error in sufficient time. The tracking speed is governed by the camera’s motor. When the error is <10% and the object is too small, the active camera scales up the object image using the zoom function with an optical lens. To effectively motion track, the speed of the object is determined by the distance travelled between sequential frames.

For motion tracking, we must predict the motion speed of the object between the current and next frame. First, the horizontal movement distance of the moving object is calculated by
(7)$$motion_{d}=\left|j_{t}-j_{t-1}\right|$$
where *j_t_* and *j_t_*_−1_ are the horizontal coordinates in the current and previous frame, respectively. Figure 8 shows the object moving between two frames. When the object is marked with a red rectangle, the red pixel (*i_t_*_−1_, *j_t_*_−1_) moves to pixel (*i_t_*, *j_t_*) in a horizontal direction. The parameter of *motion_d_* is employed to estimate the motion speed of the object.

Assuming that the object is moving at a constant speed, the camera motor is controlled to meet this speed for object tracking. If the object continues to move, we must estimate the motion distance, which can be calculated by
(8)$$motion_{est}=motion_{d}\times \#frame\;$$
*motion_est_* is the predictive motion displacement, and *#frame* is the number of frames required for tracking the object. If the estimated displacement is included in the tracking control parameters, the rotation angle and speed of the control camera can be adjusted to follow the object motion speed.

Our goal for the surveillance system is to keep the tracking target in the central area of the sampling image. In order to achieve this goal, the previously searched tracking points are used to convert the parameters to control the camera’s rotation. Thus, an active camera must be employed. The active camera can be rotated 360° horizontally and 90° vertically, and it can provide a 37× optical zoom [24]. Table 1 lists the control data for an active camera. This camera has the function of an iris for surveillance in dark environments. The zoom in/out function can be used to control the size of the tracked object. The tilt up/down function is used to rotate the camera’s vertical position with the relative control speed. Moreover, the pan left/right function is used to control the camera’s horizontal position.

The movement speed of the object is checked by the parameter with *frame_z_*, which is defined by
(9)$$frame_{z}=\{\begin{array}{c} frame_{z}+1,\;\;\mathrm{if}\;\left|j_{t}-j_{t-1}\right|\leq 15\\ 0,\;\;\mathrm{else}\; \end{array}$$

When the distance of the tracking point is less than Threshold = 15, *frame_z_* increases by one. Otherwise, *frame_z_* is set to zero. When the movement speed is low, the distance of the tracking point is lower than 15. Hence, *frame_z_* is larger, and the relative speed is lower. As the object movement speed is high, the camera cannot latch onto the object accurately. If the zoom in function is performed for an object with a high movement speed, it fails to track. To avoid this drawback, only low-motion objects are zoomed in on while continuing to track. When *frame_z_* is larger than 30, it can be ensured that this object has a low speed, and then the magnification of zooming in, dependent on the object’s size, can be performed. The object’s size is calculated with a rectangular area. If the object’s size is small, the scale of the auto zoom goes up to increase the object’s resolution and to improve recognition accuracy.

The proposed surveillance system can automatically sweep the surroundings by controlling the position of the camera. If the surroundings are partitioned into L areas, the existence of human objects can be detected with one-by-one area checking. Figure 9 shows object tracking by sweeping the surroundings with L areas. Initially, L = 0, and the camera catches the current surrounding image for object detection and stays at this position for N seconds. During this time, it can be detected whether a human object is found. If so, the surveillance system locks onto the object and tracks it for movement. Otherwise, L increases by one to check the next scene with the same procedure until it finds a human object. However, when L = Lmax, set L = 0, and the camera restarts to the initial position to continue checking the images for various regions, where Lmax is the maximum region value.

In many surveillance situations, there may be multiple objects in a single frame. The system tested here tracks each human individually. Figure 10 shows the logical process for extending this system to one that tracks multiple objects. First, each object is detected and marked with a rectangle. If only one object appears, the object is marked for tracking by controlling the camera’s motor. However, if there are multiple objects in one frame, the clearest maximum object is marked first by being assigned a color and a rectangular area. This first object is then followed, and zooming is engaged if the object is too small. The first object is followed for N seconds, and then the system searches for a secondary object. This second maximum object is marked with color 2. The camera then tracks this object for N seconds. This procedure continues for each object until all objects have a color and a rectangle. Once all objects are identified and marked, the camera returns to the initial object by recalling the relative position.

## 4. Experiments and Results

To realize the proposed surveillance system, an active camera [24] is employed, which supports the NTSC/PAL format with composite signals. This camera has the auto gain control, auto focus and auto exposure functions. The output signal is digitalized with an image catch card [25]. Figure 11 shows the setup for the proposed surveillance system. The digitalized image in a 640 × 480 format is read to a personal computer (PC). C programming is employed to implement the object detection, object tracking, and zoom in/out functions. For object tracking, the RS232 to RS485 converter is used as a bridge between the camera and PC. According to Table 1, the PC can send a command to control the position of the camera for right/left and up/down shifting. Moreover, the zoom function can be performed through this interface with a command that controls the lens of camera by enlarging the optical image.

Figure 12a shows the original image captured by the camera, and Figure 12b shows the object detection stage, where a human is successfully marked in red. After a few frames, another human enters the frame, as shown in Figure 12c. Because two humans are present, multiple object processing begins, and the new human is detected and marked in green, as shown in Figure 12d. The two objects move in opposite directions, as shown in Figure 12e. The detection results are shown in Figure 12f. In multiple object scenarios, two or more objects may overlap or pass behind or in front of one another, as shown in Figure 12g. When this happens, the objects temporarily merge until they both pass into view again, as shown in Figure 12h. Finally, the two objects fully separate again as the red object moves to the left of the frame and the green continues to the right, as shown in Figure 12i,j. The results show that the two objects can be detected accurately.

Next, the auto scan function for object tracking is experimented. The tested surroundings are partitioned into seven regions for auto sweeping to find the object, according to the processing flow of Figure 9. The horizontal and vertical angles of the camera can be controlled to sample various surrounding images. Figure 13a–e show the seven swept regions to find the object. For example, Position 1 in Figure 13a is controlled by the camera with a horizontal and vertical angle of 89.5/1.6, respectively. Once the object is detected in any position, the object is tracked by moving the position of the camera.

To demonstrate the ability of the camera to zoom effectively, it is necessary to track an object and then zoom to improve the resolution. In this situation, if an object moves too quickly, the zoom possibly fails. To ensure that the zoom works successfully, the object movement speed is checked by (7). If this condition is satisfied, the camera is be able to zoom. Figure 14 shows the result of the camera zooming in to capture a higher resolution image, where Figure 14a is the original and b–f are the same image under the following magnifications: 2×, 3×, 5×, 6× and 8×, respectively. It can be seen that, when the image of the human face is enlarged, the resolution increases as well, and it becomes easy to identify a person using a facial recognition algorithm.

Another required test of the system is for the automation of tracking with zoom for multiple objects. Figure 15a shows the original image, where two objects are detected, and both targets are marked with colored rectangles, as shown in Figure 15b. Next, the face of target 1 is enlarged using a 4× zoom for several seconds. This is sufficient time to capture a clear image of the target’s face for recognition. The camera then briefly returns to its original, neutral zoom setting and zooms in on the second target’s face for several seconds. The real-time experimental videos for multiple objects tracking and zooming in are shown in [26]. The original and enlarged images are recorded on the surveillance system’s disk. This system overcomes one common shortcoming of conventional surveillance systems, whereby human objects are often too small, making them difficult to recognize.

The proposed surveillance system can be combined with a facial recognition algorithm to identify who enters the space. Sampling images with face rectangles were selected for our facial recognition algorithm [27] for training and recognition. The test employs twenty humans as a benchmark using our proposed surveillance system, which provides various face image sizes with the zoom in function. Table 2 lists the results of various image sizes from zooming in. The results show the average of twenty tested humans. The size of the original test image is only 20 × 20 for face sampling from a far distance. Since the image is too small, it is difficult to recognize with either an algorithm or human eyes. Using a facial recognition algorithm, the testing accuracy is quite low, and in many cases, it appears as an unknown result. The proposed surveillance system can zoom in on face images to improve the resolution. The optical zoom in this method can keep the image clear to help improve facial recognition. The zoom in function employs 4×, 9× and 16× zoom for testing. As the resolution of the faces is improved, the relative accuracy can be promoted.

## 5. Discussion

There is much literature about surveillance systems, so we selected some papers for comparison, as listed in Table 3. The targets of the surveillance system include humans, cars, ships, etc. K. Garg et al. [7] presented the detection of vehicles on roads, which employed background modeling for low-complexity foreground detection. With a robust block-based method, it can overcome the problems of illumination changes, camera jitter, stationary vehicles and heavy traffic. Zhang et al. [12] presented a CNN model for human detection, especially for pedestrians on the road, and it also can recognize object attributes. Z. Shao et al. [21] proposed the detection of inshore ships with an on-land surveillance camera, which uses a new neural network for ship detection, including ship discriminative features, deep features, saliency maps and coastlines. A. Shifaf et al. [28] presented a detection and encryption method for a human surveillance system, which can encrypt faces, motion regions and background regions. The weakness of existing works is that the monitoring range is fixed in a corner. There are many dead regions using this method. Another drawback is that the tracked human object is too small. The accuracy of facial recognition is poor.

In this paper, we propose adaptive temporal processing for object detection. By controlling the active camera, the object can be tracked with 360° rotation. With auto scanning for the surroundings, the moving object can be detected and then tracked on time. With the zoom in function, a small object can be enlarged to improve its image resolution. Particularly for the facial region, the image resolution can be improved efficiently, and the recognition rate can be greatly promoted using a facial recognition algorithm. The proposed surveillance system, combined with facial recognition, can identify who enters into a space.

## 6. Conclusions

This paper proposes and implements an intelligent surveillance system with object tracking and real-time automatic zoom functionality. The system has moving object detection, moving object tracking, automatic area scanning and a zoom function. When detecting moving objects, the segmentation algorithm is used to overcome situations where objects with little to no movement are not identified. Morphological processing with expansion and contraction is successful at suppressing image noise and successfully defining the shape of objects. The tracking point for objects is used to calculate a horizontal and vertical distance relative to an image center point, and it can then govern the camera mechanism to enable centering the object in the camera’s frame. A motion estimation procedure is used to predict the speed of objects moving through the frame, and this is able to adapt to faster-moving objects by increasing the camera’s response speed. Extra rules regarding zooming while tracking are implemented to avoid object tracking failures while simultaneously zooming.

The system is able to combine object detection, tracking and zooming to improve detection resolution, which increases the probability of facial recognition success. In a test environment, this system is able to successfully detect moving objects, even when there are multiple objects to track. The camera is governed automatically by a computer and is able to track objects of different speeds and directions. For the zoom in/out experiments, according to the size of the moving object, zooming should be performed at any scale. A multi-object tracking algorithm is proposed to identify and mark moving objects with different colors so they can be distinguished and tracked separately. If deployed with some adjustments, this system would be able to scan specific areas, ignore low-activity areas, recognize moving individuals and thus deter crime.

## Figures and Tables

**Figure 1 sensors-22-08791-f001:**
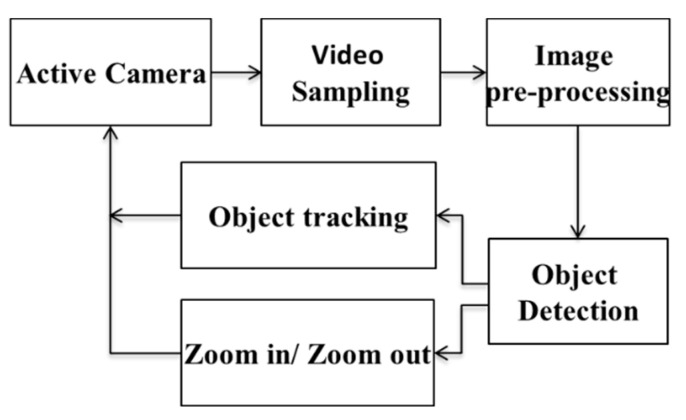
Block diagram of the proposed algorithm.

**Figure 2 sensors-22-08791-f002:**
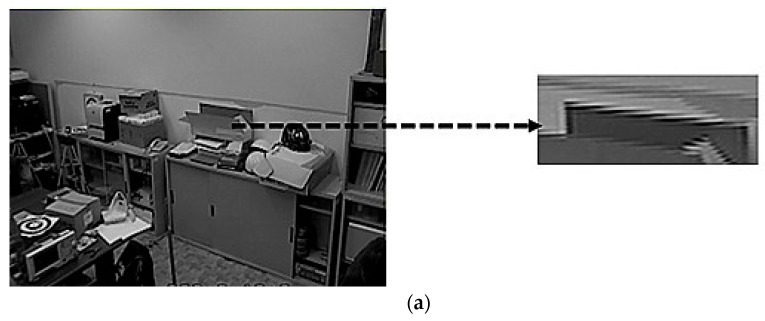
(**a**) Original sampling image; (**b**) After our median-like processing.

**Figure 3 sensors-22-08791-f003:**
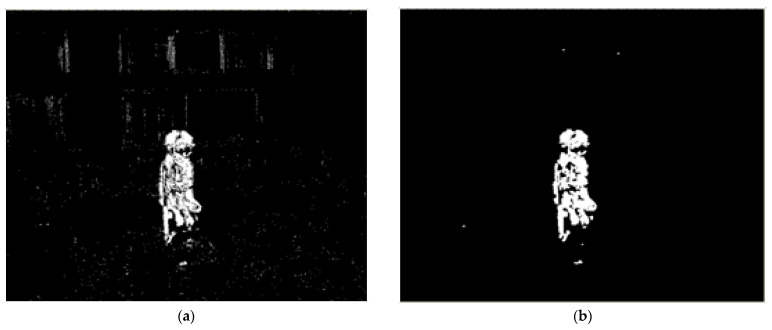
The binary image: (**a**) Original; (**b**) After erosion and dilation processing.

**Figure 4 sensors-22-08791-f004:**
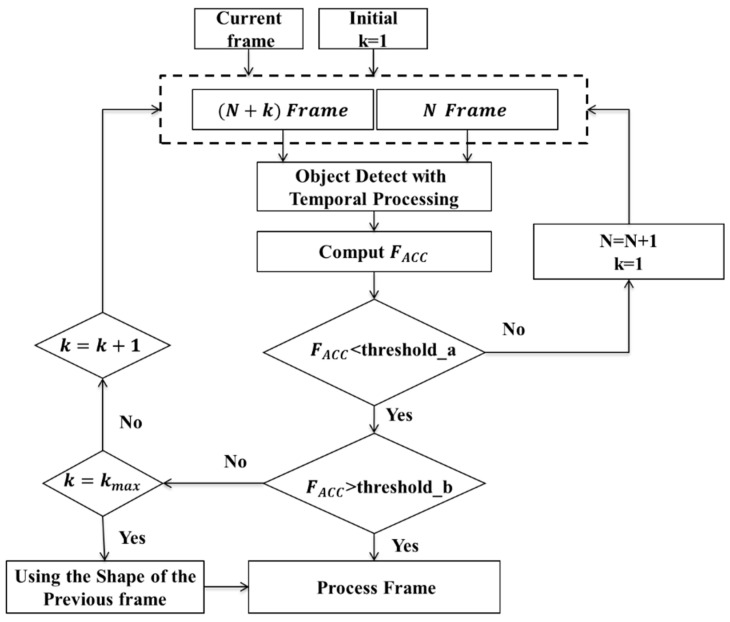
Flowchart of adaptive motion object segmentation.

**Figure 5 sensors-22-08791-f005:**
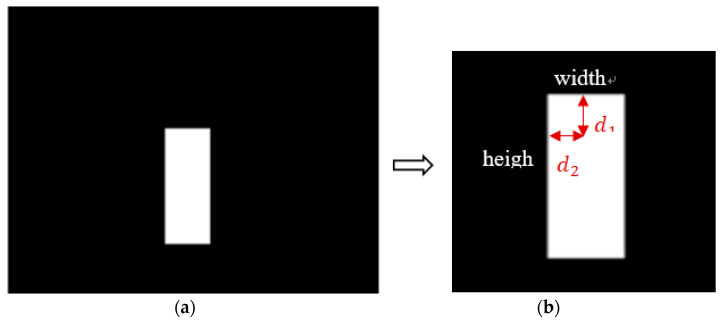
Tracking point: (**a**) Rectangle of segmented object in Figure 3b; (**b**) Head point decision.

**Figure 6 sensors-22-08791-f006:**
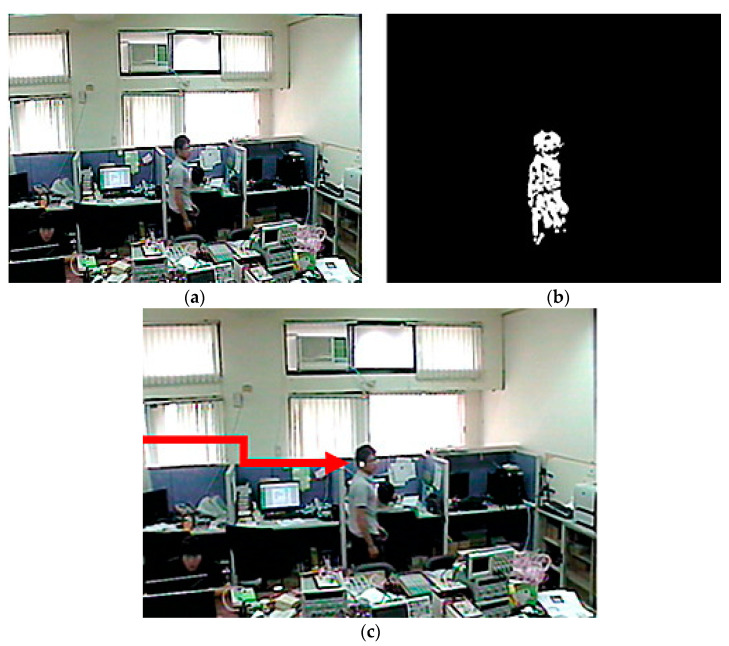
Tracking point search: (**a**) Original sample image; (**b**) Motion object shape; (**c**) Tracking point.

**Figure 7 sensors-22-08791-f007:**
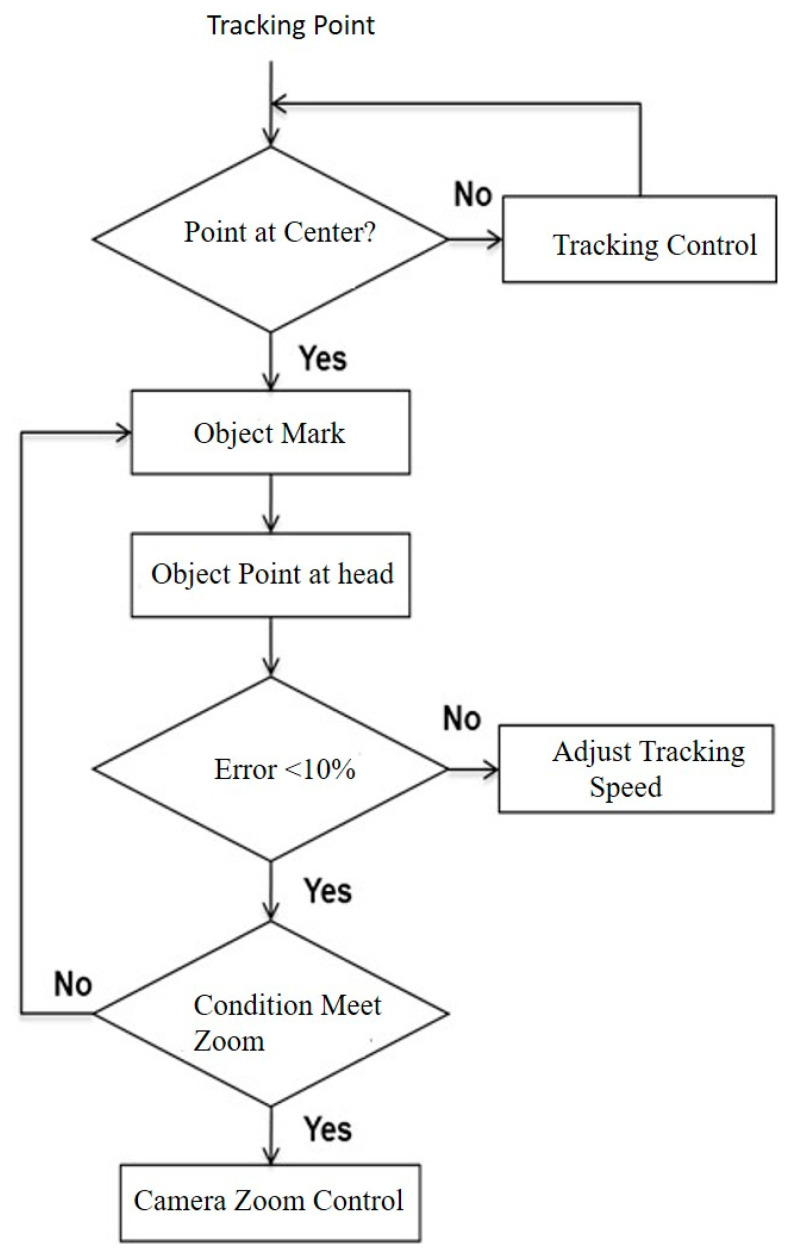
Processing flow of motion tracking control.

**Figure 8 sensors-22-08791-f008:**
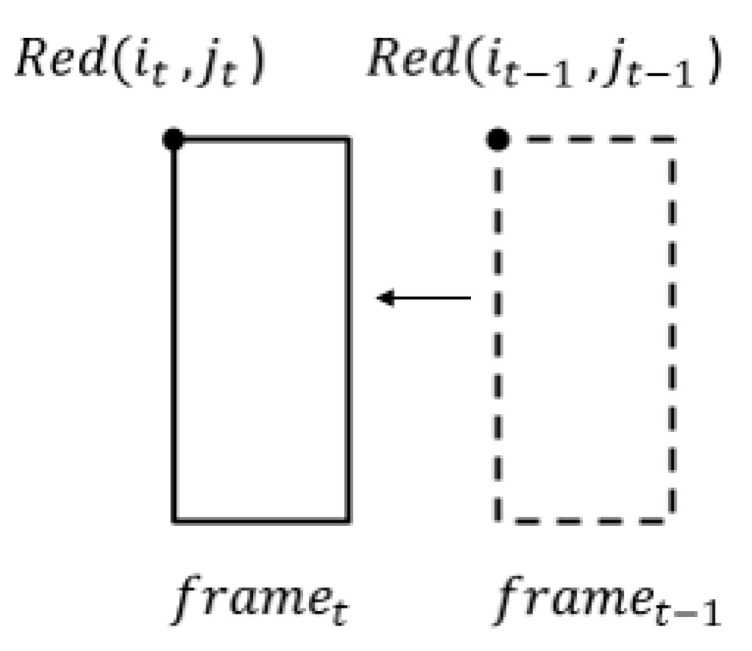
Object moving between two frames.

**Figure 9 sensors-22-08791-f009:**
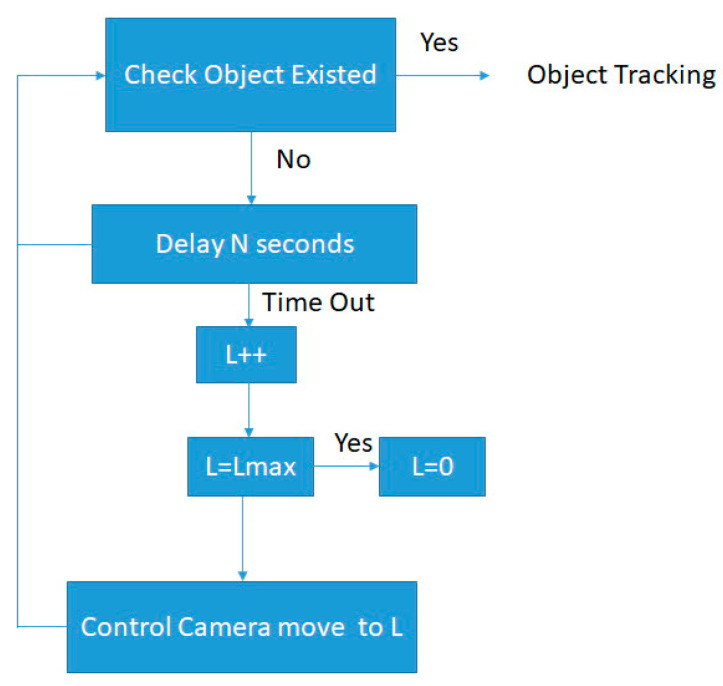
Object tracking by sweeping surroundings.

**Figure 10 sensors-22-08791-f010:**
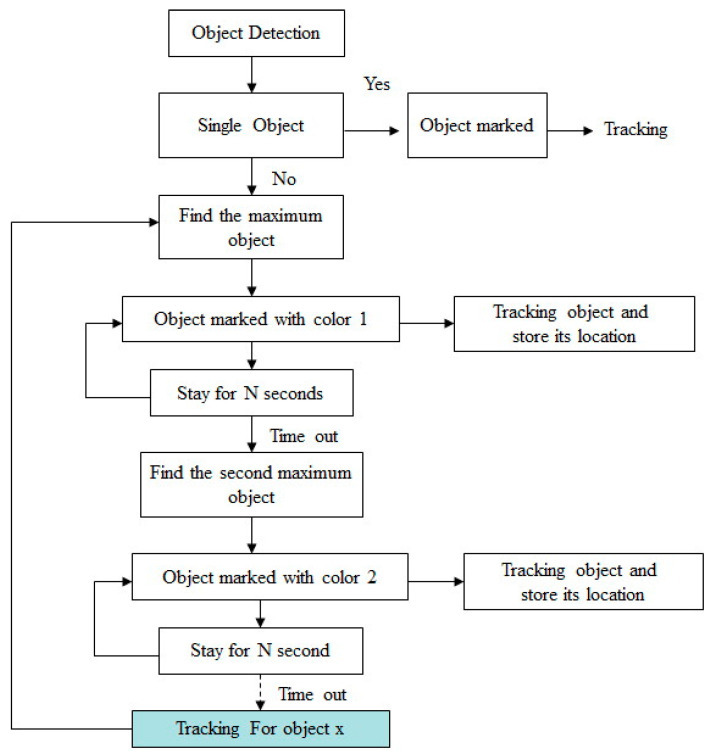
Processing flow for the multiple objects tracking.

**Figure 11 sensors-22-08791-f011:**
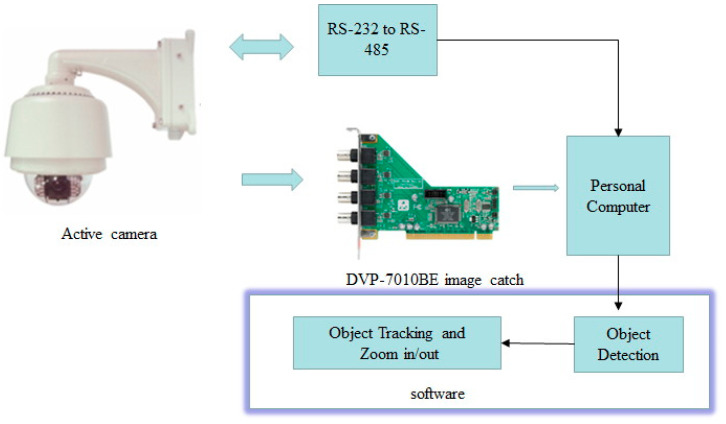
Setup for the proposed surveillance system.

**Figure 12 sensors-22-08791-f012:**
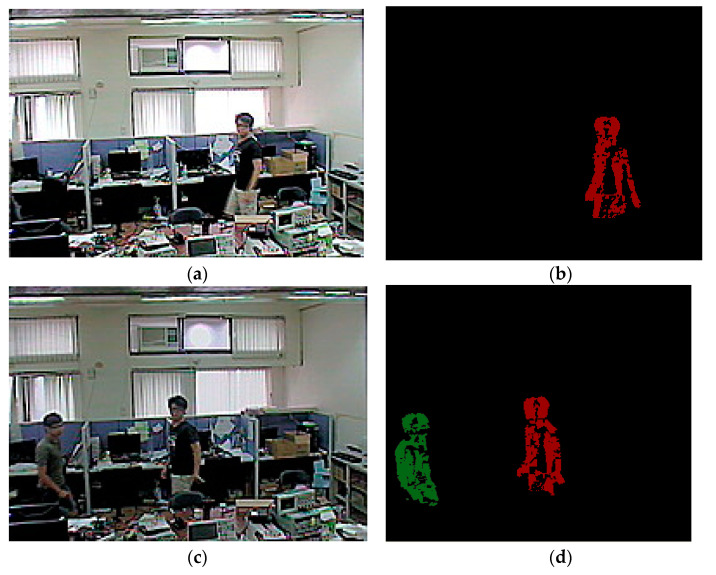
Object detection: (**a**) Original single object; (**b**) Detection result from (**a**); (**c**) Image of two objects; (**d**) Results marked with different color for (**c**); (**e**) Two objects moving; (**f**) Object segmentation from (**e**); (**g**) Two objects overlapping; (**h**) Result from (**g**); (**i**) Two objects separating; (**j**) Result from (**i**).

**Figure 13 sensors-22-08791-f013:**
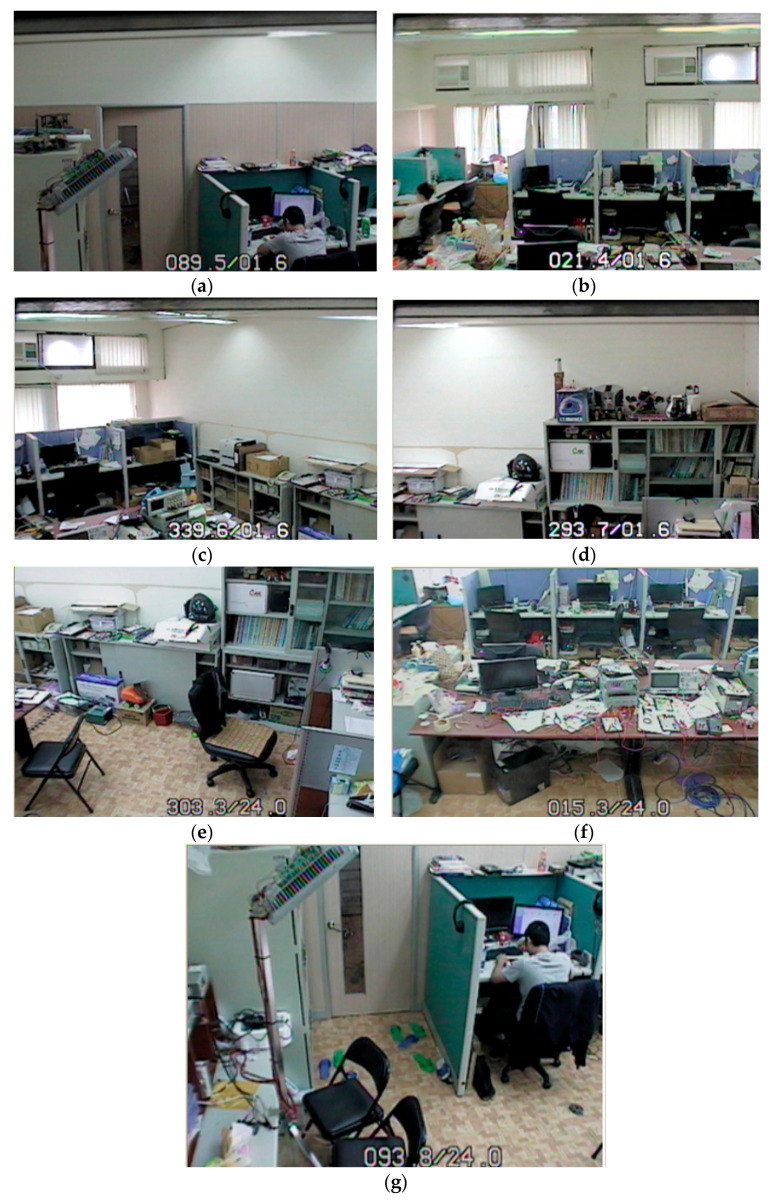
Seven swept positions: (**a**) Position 1 (89.5/1.6); (**b**) Position 2 (21.4/1.6); (**c**) Position 3 (33.9/1.6); (**d**) Position 4 (293.7/1.6); (**e**) Position 5 (303.3/24); (**f**) Position 6 (15.3/24); (**g**) Position 7 (93/24).

**Figure 14 sensors-22-08791-f014:**
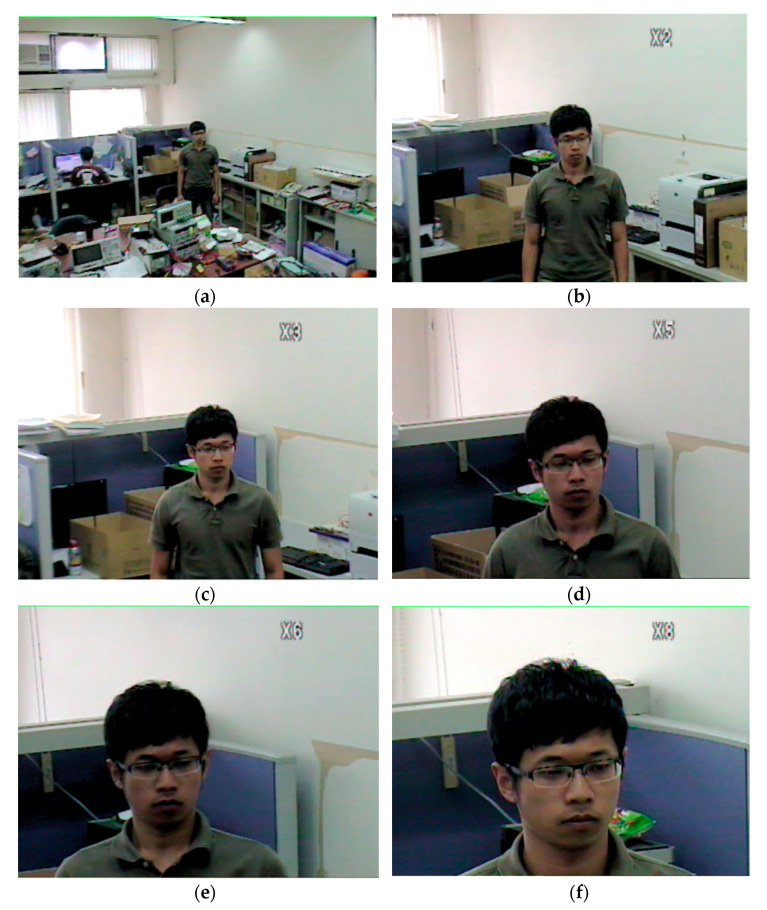
Zoom in results: (**a**) Original image; (**b**) Zoom in 2×; (**c**) Zoom in 3×; (**d**) Zoom in 5×; (**e**) Zoom in 6×; (**f**) Zoom in 8×.

**Figure 15 sensors-22-08791-f015:**
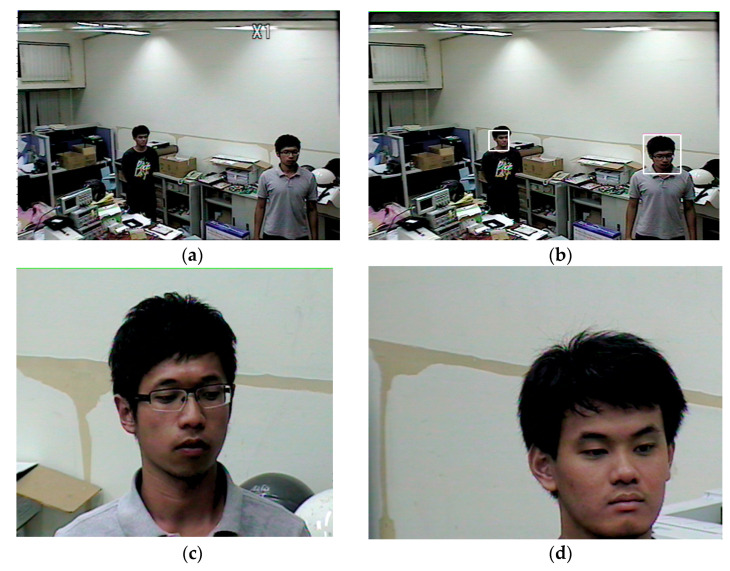
Auto-tracking combined with zooming in: (**a**) Original image (**b**) Faces marked with rectangles; (**c**) Face enlarged for object 1; (**d**) Face enlarged for object 2.

**Table 1 sensors-22-08791-t001:** Control signal for the active camera.

	**Bit Number**							
	7	6	5	4	3	2	1	0
Data Byte1	0	0	0	0	0	Iris Close	Focus Open	Focus Near
Data Byte2	Focus Far	Zoom In	Zoom Out	Tilt Down	Tilt Up	Pan Left	Pan Right	0
Data Byte3	Pan Speed 00 to 3F							
Data Byte4	Tilt Speed 00 to 3F							

**Table 2 sensors-22-08791-t002:** Facial recognition for various face sizes.

	Original Image	Zoom in 4×	Zoom in 9×	Zoom in 16×
Face Size	20 × 20	40 × 40	60 × 60	80 × 80
Accuracy	48.3%	76.5%	82.6%	91.2%
Error	26.5%	13.3	11.8	6.5%
unknown	25.2%	10.2%	5.6%	2.3%

**Table 3 sensors-22-08791-t003:** Comparison with existing surveillance systems.

	K. Garg [7]	Zhang [12]	Z. Shao [21]	A. Shifa [28]	Proposed
Target	Car	Human	Ship	Human	Human
Method	Background Modeling	Neural Network	Neural Network	Multi-Level Video Security	Adaptive Segmentation
Video Capture	Fixed	Fixed	Fixed	Fixed	360° Rotation
Function	Object Detection	Object Detection, Attribute Recognition	Object Detection	Object Detection, Encryption	Object Detection, Tracking, Recognition
Zoom	no	no	no	no	yes
Auto Scan	no	no	no	no	yes

## Data Availability

The data are openly available in a public repository that issues datasets with DOIs.

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
