# Peer review of "Intelligent Object Tracking with an Automatic Image Zoom Algorithm for a Camera Sensing Surveillance System"

_sensors, 2022, doi:10.3390/s22228791_

Round 1

Reviewer 1 Report

The content of this paper is interesting but has some minor drawbacks as below:

1. English is poor. In abstract,  "narrow morning range" ,"expand the morning range to 360"

2. In page. 6, "For tracking the human, the active camera can rotate its direction to localize the human as the object is moving and need a point to position its location in order to control the motors of camera." The control of the motors of camera may complicate the system and is not helpful for face recognition. 

Author Response

  1. English is poor. In abstract, "narrow morning range" ,"expand the morning range to 360"

Ans: In abstract, the English is revised by a professional editor.

  1. In page. 6, "For tracking the human, the active camera can rotate its direction to localize the human as the object is moving and need a point to position its location in order to control the motors of camera." The control of the motors of camera may complicate the system and is not helpful for face recognition.

Ans: The control of the motors of camera is for tracking the human moving and can zoom in the face region to increase the successful rate of face recognition.

Reviewer 2 Report

Please see the attachment:

Author Response

 Figures are enhanced and rectified throughout to improve the picture quality. We had added more explanation for the simulation results. 

Reviewer 3 Report

This paper describes the object tracking with with automatic zoom algorithm for camera surveillance systems. The comments/suggestions about paper are as follows:

1. Abstract need to be re-written as it mostly introduces the topic rather than describing what is being done in proposed work. 

2. Literature review is too short and problem statement is not clearly described. It is suggested to have a separate section of related work. Strengths and weaknesses of latest existing methods be clearly discussed and be connected to problem statement of proposed work. 

3. Please describe the working of 'Object detect with temporal processing block' in Figure with more detail.

4. Although authors have shown comparison of proposed work with 3 existing methods n Table 3. However, it seems to be not enough. The proposed work uses human as target whereas two of compared work use car and ship as target. It is suggested to compare the proposed work with others for human as target for fair comparison.

5. It is also suggested to evaluate the proposed work on some standard dataset and be compared on the same dataset with latest existing work of literature.

6. Moreover, object comparison of proposed work should also be done to demonstrate the effectiveness of proposed work.

Author Response

  1. Abstract need to be re-written as it mostly introduces the topic rather than describing what is being done in proposed work.

Ans: The abstract is re-written by a professional English editor.

  1. Literature review is too short and problem statement is not clearly described. It is suggested to have a separate section of related work. Strengths and weaknesses of latest existing methods be clearly discussed and be connected to problem statement of proposed work.

Ans: The literature review is separated with the Section 2 in the revised version. Also, we add the comments about strengths and weaknesses of existing methods.

  1. Please describe the working of 'Object detect with temporal processing block' in Figure with more detail.

Ans: The object detection is enhanced in the revised version.

  1. Although authors have shown comparison of proposed work with 3 existing methods n Table 3. However, it seems to be not enough. The proposed work uses human as target whereas two of compared work use car and ship as target. It is suggested to compare the proposed work with others for human as target for fair comparison.

Ans: We had added the anther paper [12] for comparisons in Table 3, which is for human detection and attribute recognition.

  1. It is also suggested to evaluate the proposed work on some standard dataset and be compared on the same dataset with latest existing work of literature.

Ans: The proposed techniques must use the active camera to prove its function and real-time detection accuracy, which cannot be estimated by using dataset.

  1. Moreover, object comparison of proposed work should also be done to demonstrate the effectiveness of proposed work

Ans: We had added the demonstrated video following Web.

http://www.youtube.com/watch?v=iSC8VFOV0zc&feature=youtu.be

Round 2

Reviewer 3 Report

The authors have revised the manscript mostly satisfactorily as per comments mentioned by me in last cycle of review. However, there is need to connect the weaknesses of existing work with problem statement in more elaborative way.

Author Response

The revised paper had addressed" the weaknesses of existing work with problem statement" in page 18.